# Stage I Squamous Cell Carcinoma of the Anus: Is Radiation Therapy Alone Sufficient Treatment?

**DOI:** 10.3390/cancers12113248

**Published:** 2020-11-04

**Authors:** Eric Miller, Ansel Nalin, Dayssy Diaz Pardo, Andrea Arnett, Laith Abushahin, Syed Husain, Ning Jin, Terence Williams, Jose Bazan

**Affiliations:** 1Department of Radiation Oncology at the Arthur G. James Cancer Hospital and Richard J. Solove Research Institute, The Ohio State University Comprehensive Cancer Center, Columbus, OH 43210, USA; eric.miller@osumc.edu (E.M.); ansel.nalin@osumc.edu (A.N.); dayssy.diazpardo@osumc.edu (D.D.P.); andrea.arnett@osumc.edu (A.A.); terence.williams@osumc.edu (T.W.); 2Department of Internal Medicine, Division of Medical Oncology at the Arthur G. James Cancer Hospital and Richard J. Solove Research Institute, The Ohio State University Comprehensive Cancer Center, Columbus, OH 43210, USA; laith.abushahin@osumc.edu (L.A.); ning.jin2@osumc.edu (N.J.); 3Colon and Rectal Surgery at the Arthur G. James Cancer Hospital and Richard J. Solove Research Institute, The Ohio State University Comprehensive Cancer Center, Columbus, OH 43210, USA; syed.husain@osumc.edu

**Keywords:** anal cancer, stage I, chemoradiation

## Abstract

**Simple Summary:**

The optimal treatment of early stage anal cancer is unknown. This patient population was relatively under-represented on the trials, which defined radiation therapy with concurrent chemotherapy as the standard treatment for anal cancer, thus radiation therapy alone may be an effective alternative treatment. The aim of this study was to use a large national database of anal cancer patients to compare overall survival of patients treated with radiation therapy alone to those treated with radiation therapy with concurrent chemotherapy. We found that patients who received radiation therapy alone were more likely to be ≥70 years old and less likely to be female. Treatment with radiation and concurrent chemotherapy was associated with a 31% reduction in the risk of death compared to treatment with radiation alone. Our results suggest that radiation with concurrent chemotherapy should be the standard treatment for early stage anal cancer patients.

**Abstract:**

The optimal treatment for stage I squamous cell carcinoma of the anus (SCCA) remains undefined. Recently, wide local excision alone was found to have comparable survival to those treated with chemoradiation (CRT). Given that local excision may be sufficient for the treatment of stage I SCCA, we hypothesized that radiation therapy (RT) alone, compared to CRT would result in equivalent overall survival (OS) in this population. We identified non-surgically treated patients with stage I SCCA from the National Cancer Database from 2004–2015. We included only patients treated either with CRT (45–59.4 Gy with chemotherapy initiated within 14 days of RT) or RT alone (45–59.4 Gy with no chemotherapy). The primary endpoint was OS between CRT and RT patients. Propensity-score matched (PSM) analysis was performed to determine the effect of concurrent chemotherapy on OS using a Cox proportional hazards model with robust standard error to account for clustering in matched pairs. We identified 3552 stage I patients treated with CRT and 287 treated with RT. Compared to patients treated with CRT, those that received RT were more likely to be ≥70 years old (33.1% vs. 19.7%, *p* < 0.001) and less likely to be female (63.1% vs. 71.0%, *p* < 0.001). The proportion of patients with a Charlson-Deyo score of 0 was similar in both groups (80.8% RT vs. 82.7% CRT, *p* = 0.164). The PSM cohort consisted of 287 pairs of patients with median follow-up 48.3 months (interquartile range, 24.4–85.1 months) and 151 deaths (86 RT, 65 CRT). CRT was associated with a 31% reduction in the risk of death (HR = 0.69, 95% CI 0.50–0.95, *p* = 0.023). We found that CRT was associated with improved OS, compared to RT alone, in patients with non-surgically treated stage I SCCA. These data suggest that de-intensification of therapy in stage I SCCA must be used with caution. However, given the retrospective nature of the data, prospective trials are required.

## 1. Introduction

Concurrent chemoradiation (CRT) remains the standard of care treatment for squamous cell carcinoma of the anal canal (SCCA) based on two large randomized trials (United Kingdom Coordinating Committee for Cancer Research [UKCCCR] ACT I trial and the European Organisation for the Research and Treatment of Cancer [EORTC] trial) that compared CRT to radiation therapy (RT) alone [1,2,3]. The National Comprehensive Cancer Center Network Guidelines recommend CRT for all patients with localized SCCA [4]. However, the trials that demonstrated the superiority of CRT over RT included relatively few patients with small (≤2 cm), node-negative (T1N0) tumors. Patients with T1N0 disease represented <15% of the population on the UKCCCR ACT I trial and 0% on the EORTC trial. Subsequent studies looking at induction chemotherapy [5], eliminating mitomycin C [6], and substituting cisplatin for mitomycin C [7] have also only included a small proportion of patients with T1N0 disease.

Given that patients with T1N0 SCCA were relatively under-represented or excluded from the randomized trials, questions remain as to whether CRT, which is associated with significant acute and late morbidities, is necessary. Subset analysis from the ACT I trial did demonstrate a significant benefit of CRT over RT for local control in patients with T1-2N0 disease (RR = 0.45, 95% CI 0.29–0.71, *p* = 0.0005) [8]. In addition, several retrospective series have evaluated CRT versus RT alone in patients with T1-2N0 SCCA and have found favorable results in patients that received RT alone [9,10,11]. A recent study of patients with T1N0 SCCA from the National Cancer Database (NCDB) found no difference in overall survival (OS) between 503 patients treated with local excision alone compared to 1740 patients treated with CRT [12].

Here, we first set to evaluate factors associated with receipt of RT alone versus CRT in patients with T1N0 SCCA using the NCDB. We next evaluate the hypothesis that since local excision without any adjuvant treatment may be sufficient treatment for patients with T1N0 SCCA, RT alone should result in equivalent OS compared to CRT in this favorable subgroup of patients.

## 2. Results

### 2.1. Baseline Characteristics

We identified 3552 patients treated with CRT and 287 patients treated with RT that met the study criteria (Figure 1). Patient characteristics for the entire cohort are shown in Table 1. Overall, patients treated with RT were more likely to be ≥70 (33.1% vs. 19.7%, *p* < 0.001), less likely to be female (63.1% vs. 71.0%, *p* < 0.001) and less likely to be of white race (87.5% vs. 90.9%, *p* = 0.054) compared to patients in the CRT group. In addition, there was an uneven distribution of tumors by size in the RT and CRT groups, with larger tumors in 44.6% versus 53.9% (*p* = 0.009), but the size was unknown in 28.6% and 23.0% of patients in the RT, and CRT groups, respectively. The proportion of patients with a Charlson-Deyo score of 0 was similar between the RT and CRT groups (80.8% vs. 82.7%, *p* = 0.164). Finally, duration of radiation therapy was similar in the RT alone and CRT groups (51.5 days vs. 54.4 days, *p* = 0.615).

### 2.2. Factors Associated with RT Alone versus CRT

Table 2 demonstrates the univariate and multivariate logistic regression analyses for factors associated with receiving RT versus CRT. Patients ≥70 years old were more likely to receive RT alone than CRT when adjusted for other covariates (OR = 2.45, 95% CI 1.76–3.39, *p* < 0.001). In addition, female patients and patients with tumors >1–2 cm (versus ≤1 cm) were more likely to receive CRT.

### 2.3. Overall Survival in CRT and RT Patients

The median follow-up for the entire cohort was 51.9 months (interquartile range [IQR], 26.0–85.8 months). At the time of analysis, 697 deaths had occurred (611 in the CRT group and 86 in the RT alone group). The estimated 4-year OS was 87.1% in the CRT group versus 75.7% in the RT alone group (*p* < 0.001) (Figure 2). Table 3 demonstrates the results of the univariate and multivariate Cox proportional hazards regression analysis in the entire cohort. On multivariate analysis, treatment with CRT was associated with a 35% reduction in the risk of death (HR = 0.65, 95% CI 0.52–0.82, *p* < 0.001). Female gender was associated with improved OS (HR = 0.69, *p* < 0.001) while age 60–69 (HR = 1.53, *p* < 0.001) and ≥70 years (HR = 3.65, *p* < 0.001), Charlson-Deyo score of 1 (HR = 1.50, *p* < 0.001) or 2–3 (HR = 2.77, *p* < 0.001), and living in an area with median household income <$46,000/year (HR = 1.27, *p* = 0.003) were all associated with worse OS.

In order to minimize the bias from potential confounders, we next performed a propensity-score based analysis. In the propensity-score matching, a CRT match (using a caliper size of 0.006 based on a propensity score SD of 0.029) was identified for 287 of the possible 287 patients treated with RT alone. The distribution of propensity scores was similar between the CRT and RT groups (Appendix A
Appendix A). All baseline covariates were well-balanced between the CRT and RT groups based on a standardized difference of <0.10 (Appendix A
Appendix A). The median follow-up for the 574 patients in the PSM cohort was 48.3 months (IQR, 24.4–85.1 months) at which point 151 patients had died (65 in the CRT group; 86 RT alone group). The 4-year OS was significantly higher in patients that received CRT (84.0% versus 75.7%, HR = 0.69, 95% CI 0.50–0.95, *p* = 0.023) (Figure 3). As an exploratory analysis, we compared outcomes of patients treated with CRT to those undergoing wide local excision using propensity-score matching (distribution of scores shown in Appendix A
Appendix A, standardized differences shown in Appendix A
Appendix A). We found that there was a slightly higher risk of death with wide local excision compared to CRT (4-year OS 82.8% vs. 85.6%, HR = 1.20, 95% CI 1.00–1.43, *p* = 0.045) as shown in Appendix A
Appendix A.

## 3. Discussion

In summary, we found that age ≥70 years, male gender, and tumors ≤1 cm were associated with a higher likelihood of treatment with RT alone compared to CRT in patients with T1N0 SCCA. In addition, we found that CRT remained associated with improved OS compared to RT alone in the entire cohort, and in our propensity-matched cohort, with a >30% relative reduction in the risk of death. To our knowledge, this is the largest study of patients with T1N0 SCCA including the largest sample of T1N0 patients that received RT only, and our findings suggest that de-escalation of standard of care CRT to RT should be done with great caution in this group of patients.

In the ACT I trial, which helped define CRT as the standard of care, a subset analysis of patients with T1-2N0 disease demonstrated a local control benefit in patients receiving CRT compared to those treated with RT alone [8]. However, the hypothesis that RT alone may be sufficient for small node-negative SCCA, has been tested in several small retrospective studies. Zilli et al. evaluated 146 patients with T1-2N0 SCCA treated with RT alone (N = 71) or CRT (N = 75) and found that on multivariate analysis, local-regional control and cancer-specific survival tended to be superior in patients treated with CRT and in those with T1N0 disease [11]. However, the number of patients with T1N0 (N = 29) disease was too small to make any definitive conclusions regarding the optimal treatment regimen for this subgroup, though 21 of these 29 patients were treated with RT alone. In a study by Ortholan et al. of 69 patients with Tis-T1 disease (all invasive tumors ≤1 cm), 3 of the 12 patients with Tis disease received excision alone and the remaining 66 patients received RT alone. These patients had favorable 5-year OS and disease-free survival rates of 94%, and 89%, respectively [9].

Further rationale for testing a treatment strategy of RT alone in T1N0 SCCA is supported by the surgical literature. As previously mentioned, Chai et al. found no difference in 5-year OS between 503 patients treated with local excision or the 1740 patients treated with CRT (85.3% vs. 86.8%) [12]. These results were similar for patients with tumors ≤1 cm (88.5% vs. 91.6%) and >1–2 cm (86.6% vs. 86.4%). In a significantly smaller study of 57 patients with stage I SCCA treated at the Mayo Clinic with either local excision (N = 13) or CRT (N = 44), the 5-year progression-free survival (PFS) rates were similar between the 2 cohorts at 91% excision vs. 83% CRT (*p* = 0.57) [13].

However, more recently, this hypothesis was called into question by an NCDB study of T1-2N0 patients that were ≤70 years old [10]. In this study, Youssef et al. found that CRT was associated with improved OS in these patients with early-stage disease. Approximately 1/3 of the patients in that study had T1N0 disease of which 1428 received CRT and 101 received RT alone. In this subgroup, the authors found a trend towards improved OS in the CRT group that did not reach significance (5-year OS 90.3% CRT vs. 84.7% RT, *p* = 0.11). Therefore, it appears that the benefit of CRT over RT in this study was driven primarily by the patients with T2N0 disease. In contrast, our study includes 2.5-fold higher number of patients (2.8-fold higher number of patients treated with RT alone) and with this larger sample size, we found that CRT is associated with improved OS. While this finding conflicts with our original hypothesis, our results suggest that CRT should still be considered for patients with T1N0 disease.

Similar to a prior study, we found the factors associated with RT were older age, male gender, and tumors ≤1 cm in size [14]. The factor most strongly associated with the receipt of RT alone was age ≥70 years. These patients were more than 2-fold more likely to receive RT alone compared to CRT. Elderly patients are also more likely to have comorbidities and both patient age and the presence of comorbidities were associated with worse OS on multivariate analysis. Despite controlling for these factors on multivariate analysis and despite the fact that the proportion of elderly patients and those with comorbidities were identical in the propensity-matched cohort, the protective effect of CRT persisted. This suggests that the protective effect of CRT cannot solely be explained by the older age of the patients in the RT cohort. Neither the ACT I trial nor the EORTC trial demonstrated an improvement in OS when comparing CRT to RT alone in the entire cohort [1,2,3]. In our patient population, we observed a nearly 8% difference in OS in favor of patients who received CRT. While, this dramatic difference in OS may be secondary to other unaccounted for, or unmeasured confounders, such as performance status as well as eligibility for further therapy at time of recurrence, the NCDB is limited in the amount of information available regarding patient performance status and captures treatment information only at disease presentation, but not further lines of therapy at the time of disease progression. Similarly, the results of our exploratory analysis, which favored improved OS for patients treated with CRT compared to wide local excision alone may be due to missing patient factors. Compared to the prior study by Chai et al., we utilized strict definitions of CRT, had a larger sample size, and used propensity score matching to help reduce the effects of confounders on our results [12]. Overall, we view the results of both our study and that of Chai et al. as hypothesis-generating in need of validation in prospective trials. In light of limited prospective data providing guidance on this treatment population, we favor treatment with CRT in patients who are eligible based on our results.

Definitive chemoradiation for anal cancer is associated with significant acute and long-term toxicity. The ACT I trial demonstrated a significant increase in acute toxicity in the CRT arm compared to RT alone, but no significant difference in late toxicity [1]. A prospective study by Han et al. assessed toxicity and quality of life (QOL) in patients with anal or perianal cancer treated with intensity modulated radiation therapy (IMRT) with concurrent chemotherapy [15]. Significant acute toxicities were reported including grade 3+ dermatologic (46% of patients), hematologic (38%), and gastrointestinal (9%) toxicities. The QOL scores were significantly worse at the end of treatment, but returned to baseline 3 months after treatment. Additional studies have focused on late toxicity following treatment with definitive CRT [16,17,18,19]. Das et al. surveyed patients who completed RT or CRT for SCC with a minimum 2-year interval since completion of treatment [17]. Nearly all (97%) patients received concurrent chemotherapy. Patients reported significant issues with diarrhea, difficulty with bowel control, and sexual activity. Similar late toxicities were reported by long-term survivors in a survey study from Norway [16]. Patients with early stage SCCA treated with IMRT have been shown to have more favorable toxicity profiles when compared to those with more advanced disease requiring higher radiation therapy doses [20]. In a retrospective review of 43 patients with cT1-2N0 SCCA treated with IMRT using a simultaneous integrated boost technique, the rates of reported acute grade 3 dermatologic, gastrointestinal, genitourinary, leukopenia, neutropenia, and thrombocytopenia were 18%, 0%, 3%, 26%, 15%, and 9%, respectively [20]. No non-heme-related grade 4 toxicities were reported. Due to the potentially severe and late toxicity of definitive treatment and the overall favorable prognosis of these patients, efforts to reduce toxicity have been made as treatment of SCCA has evolved. 

Additional de-escalation strategies exist in addition to omitting concurrent systemic therapy. White et al. reviewed patients with SCCA treated with definitive CRT comparing outcomes in patients receiving 2 doses of concurrent mitomycin-C to those receiving only 1 dose [21]. There was no difference in PFS, cancer-specific survival, colostomy-free survival, and OS between the groups with more acute grade ≥2 toxicities observed in the group receiving 2 doses of mitomycin C. Replacing concurrent infusional 5-FU with capecitabine has also been shown to reduce both grade ≥3 hematologic toxicity and treatment breaks based on a retrospective series from Goodman et al. [22]. While the results of this study highlight the importance of delivering concurrent systemic therapy, de-escalation can also be performed by altering the radiation therapy plan. Small retrospective series have demonstrated favorable outcomes following omission of prophylactic inguinal nodal irradiation in patients without clinical involvement of this nodal chain [23,24]. In addition, to reduce normal tissue toxicity, adjustments to the radiation dose delivered to target volumes can also be made. Indeed, several retrospective series have demonstrated favorable outcomes when treating patients with low volume or early stage SCCA with chemotherapy and reduced doses of RT [25,26,27]. Notably, the question of de-escalation is the subject of several current prospective clinical trials. DECREASE is a clinical trial evaluating lower dose CRT in early stage SCCA (T1-2N0M0) and is currently accruing through the Eastern Cooperative Oncology Group (NCT04166318). The PLATO (personalising anal cancer radiotherapy dose) umbrella trial (ISRCTN88455282) is funded by Cancer Research UK and includes clinical trials ACT 3 and 4. ACT 3 is a non-randomized phase II study evaluating local excision with selective postoperative CRT for patients with T1N0 anal margin tumors while ACT 4 is a randomized phase II trial comparing reduced dose to standard dose CRT for patients with T1-2 (<4 cm) N0 SCCA.

Our study has several limitations. Most notably, the NCDB does not capture data on local-regional recurrence and/or cancer-specific survival. Therefore, it is unclear if the higher mortality seen in the RT group is truly due to recurrent disease. While we were able to control and match for known covariates included in the NCDB, other unaccounted for selection biases that influence chemotherapy omission likely persist and can contribute to and/or exaggerate the difference in OS seen between the RT and CRT groups. While the Charlson-Deyo score provides some insight on patient comorbidities, it is not able to fully capture patient performance status which is a major limitation in utilizing the NCDB. In addition, the NCDB does not give details on the specific chemotherapy agents used, but nearly 90% of the patients in the CRT group received multi-agent chemotherapy. It would be interesting to compare RT alone to single-agent CRT in stage I SCCA, but since only 29 patients in the matched CRT cohort received single-agent chemotherapy, a statistical comparison would not be valid.

## 4. Materials and Methods

We performed a retrospective review using the NCDB, which is a joint project of the Commission on Cancer (CoC) of the American College of Surgeons and the American Cancer Society. The NCDB is a nationwide hospital-based database that contains de-identified hospital registry data from more than 1500 accredited facilities and represents more than 70% of newly diagnosed cancer cases in the United States [28]. The NCDB collects data on patient demographics and comorbidities, tumor characteristics and staging details, primary therapies administered, and overall survival, but not disease recurrence or salvage treatment. The CoC’s NCDB and the hospitals participating in the CoC NCDB are the source of the de-identified data and have not been verified, and are not responsible for the statistical validity of the data analysis nor the conclusions presented in this study.

Patients diagnosed with stage I SCCA from 2004–2015 were collected from the NCDB participant user file with additional inclusion and exclusion criteria summarized in Figure 1. We excluded patients that received only a surgical procedure without RT or CRT. Additional exclusion criteria included non-squamous histology, patients with unknown radiation and chemotherapy treatment details, and patients who received radiation therapy doses >59.4 Gy or <45 Gy.

### 4.1. Treatment Definitions

We defined 2 groups of patients based on treatment delivered: concurrent CRT and RT alone. Patients were considered to have received CRT if: (1) Chemotherapy was delivered within 14 days of the initiation of RT, and patients (2) received a radiation dose of ≥45 Gy but ≤59.4 Gy. This 14 day window was based on the criteria for discontinuing protocol treatment on RTOG 0529, while a minimal dose of 45 Gy is recommended in the NCCN guidelines as the minimal dose delivered to early stage disease in the definitive setting [4]. Patients in the RT alone group were required to have received ≥45 Gy but ≤59.4 Gy without any use of chemotherapy.

In an exploratory analysis conducted based on the results of our primary endpoint comparing OS in patients treated with concurrent CRT versus RT alone, we further analyzed outcomes in patients treated with excision alone compared to concurrent CRT. Excision alone was defined as local tumor excision, not otherwise specified, or excisional biopsy. Patients that received further chemotherapy, RT, or concurrent CRT were excluded from the surgery only cohort.

### 4.2. Study Variables

Categorical baseline covariates included: gender (male vs. female), race (white vs. non-white), median income (≥$46,000 vs. <$46,000), insurance status (insured vs. uninsured), facility type (academic vs. other), tumor size (≤1 cm, 1–2 cm, and unknown). Age was analyzed in 4 groups: <50 years-old, 50–59 years-old, 60–69 years-old, and ≥70 years-old. The Charlson-Deyo score, a measure of comorbidity, was categorized into 3 groups: 0 (no comorbidities), 1 (1 comorbidity), and 2–3 (>1 comorbidities). Distance to the nearest facility was analyzed as a continuous variable.

### 4.3. Statistical Methods

The objectives of this study were to determine the factors associated with receiving RT alone (versus CRT) and to evaluate OS in patients treated with CRT compared to RT alone. The differences in treatment groups between the CRT and RT groups were tested using the χ2 test (categorical variables) and the t-test (continuous variables), and logistic regression was used to identify predictors of RT vs. CRT. Variables with *p* ≤ 0.10 on univariate analysis were included in the multivariate logistic regression model. We hypothesized that elderly patients and those with comorbidities would more likely receive RT alone compared to CRT.

We evaluated OS in the CRT and RT patients. We hypothesized that, after controlling for potential confounders, CRT and RT would result in equivalent OS. We evaluated OS by the Kaplan-Meier method and by a multivariate Cox proportional hazards model (including all variables with *p* < 0.20 on univariate analysis). In order to further minimize the effect of potential confounders, we additionally used a propensity-score matched analysis. All baseline covariates mentioned in the Study Variables section were included in the propensity score model. Patients receiving RT alone were matched with those receiving CRT using a 1:1 nearest available neighbor match without replacement [29] using a caliper size calculated as 20% of the standard deviation of the propensity score [30]. Common support for the propensity score distributions was evaluated graphically and balance was evaluated by computing the standardized difference of the covariates across the two groups [31]. Following PSM, OS between treatment groups was estimated using the Kaplan-Meier method and the effect of CRT was evaluated with a Cox proportional hazards model with robust standard errors to account for clustering in matched pairs. Given the results we found comparing CRT to RT alone, we conducted an exploratory analysis of OS in patients treated with CRT versus excision alone, using the propensity-score matched analysis procedure described above. All statistical analyses were performed using SAS, version 9.4 (SAS Institute Inc., Cary, NC, USA).

## 5. Conclusions

In conclusion, we found that CRT was associated with improved OS compared to RT alone in patients with non-surgically treated stage I SCCA. These hypothesis-generating data highlight that efforts to de-escalate therapy should be performed with great caution and in the context of well-designed, prospective protocols.

## Figures and Tables

**Figure 1 cancers-12-03248-f001:**
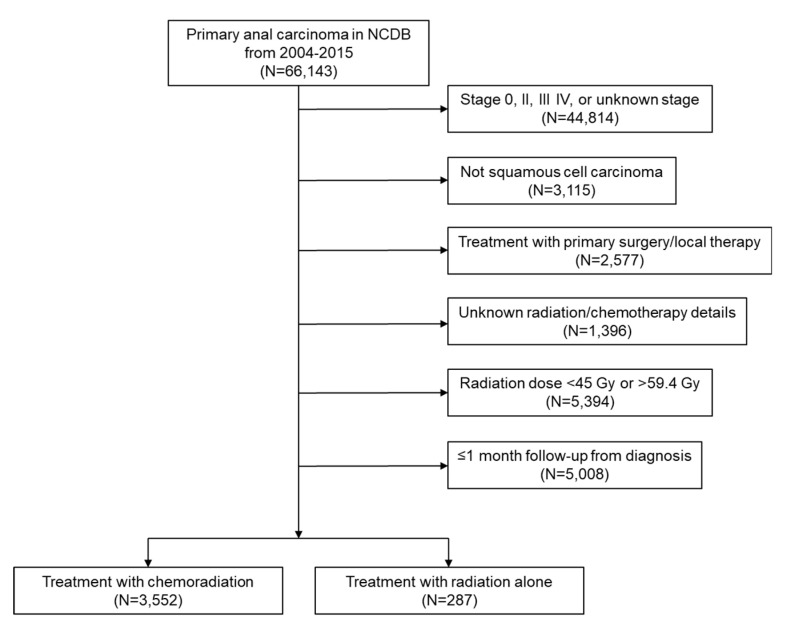
Study flow diagram for analytic cohorts. Abbreviation: NCDB, National Cancer Database.

**Figure 2 cancers-12-03248-f002:**
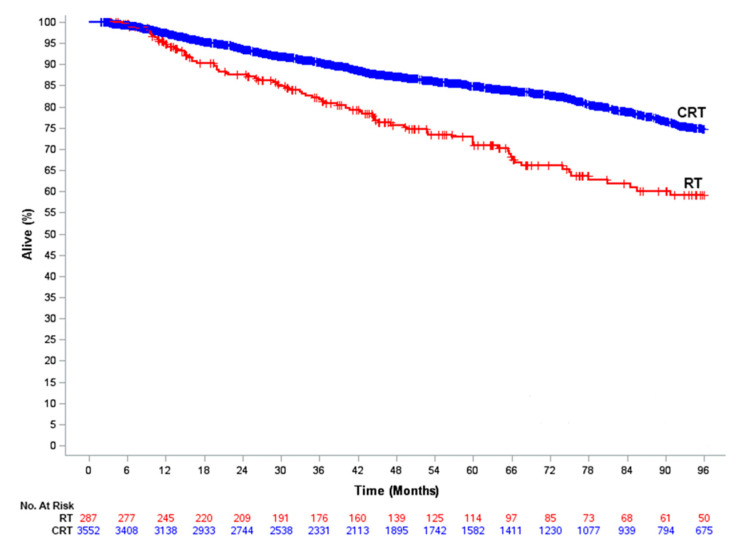
Overall survival of patients receiving definitive chemoradiation (CRT) compared to those receiving radiation therapy (RT) alone in the entire cohort. Curves represent actual survival as estimated by the Kaplan-Meier method.

**Figure 3 cancers-12-03248-f003:**
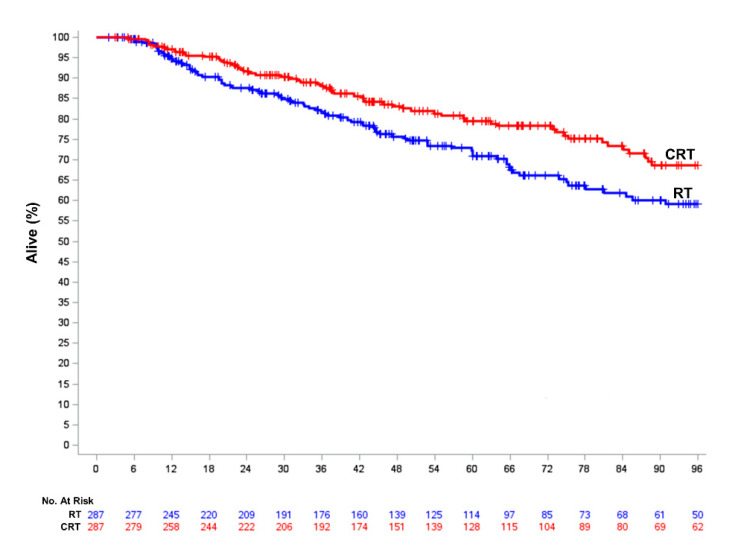
Overall survival of propensity score-matched patients treated with definitive chemoradiation (CRT) compared to those receiving radiation therapy (RT) alone. Curves represent actual survival as estimated by the Kaplan-Meier method.

**Table 1 cancers-12-03248-t001:** Patient characteristics in the chemoradiation and radiation therapy alone cohorts.

Characteristic	Chemoradiation (N = 3552), Number (%)	Radiation Therapy Alone (N = 287), Number (%)	*p*-Value
Age (years)			<0.001
<50	642 (18.1)	59 (20.6)	
50–59	1196 (33.7)	68 (23.7)	
60–69	1013 (28.5)	65 (22.6)	
≥70	701 (19.7)	95 (33.1)	
Gender			<0.001
Female	2522 (71.0)	181 (63.1)	
Male	1030 (29.0)	106 (36.9)	
Race			0.054
White	3229 (90.9)	251 (87.5)	
Non*–*white	323 (9.1)	36 (12.5)	
Charlson*–*Deyo score			0.164
0	2938 (82.7)	232 (80.8)	
1 2*–*3	403 (11.4) 211 (5.9)	30 (10.5) 25 (8.7)	
Median income			0.145
≥$46,000	1405 (39.6)	101 (35.2)	
<$46,000	2147 (60.4)	186 (64.8)	
Insurance Status			0.775
Insured	3451 (97.2)	278 (96.9)	
Uninsured	101 (2.8)	9 (3.1)	
Distance to facility, mi.	23.5 (SD = 102.3)	19.8 (SD = 38.5)	0.550
Facility type			0.459
Academic	1004 (28.3)	87 (30.3)	
Non*–*academic	2548 (71.7)	200 (69.7)	
Tumor Size			0.009
≤1 cm	821 (23.1)	77 (26.8)	
>1*–*2 cm	1914 (53.9)	128 (44.6)	
Unknown size	817 (23.0)	82 (28.6)	

mi.: miles. SD: standard deviation.

**Table 2 cancers-12-03248-t002:** Univariate and multivariate logistic regression analysis of factors associated with receiving radiation therapy alone vs. chemoradiation.

RT Alone vs. CRT	Univariate Analysis			Multivariate Analysis		
Variable	OR	95% CI	*p*-Value	OR	95% CI	*p*-Value
Age (years)						
<50	1.62	1.13–2.32	0.009	1.42	0.98–2.05	0.064
50–59	Ref	Ref	Ref	Ref	Ref	Ref
60–69	1.13	0.80–1.60	0.498	1.15	0.81–1.63	0.437
≥70	2.38	1.72–3.30	<0.001	2.45	1.76–3.39	<0.001
Female vs. male	0.70	0.54*–*0.90	0.005	0.76	0.58*–*0.98	0.036
Charlson–Deyo score						
0	Ref	Ref	Ref	Ref	Ref	Ref
1	0.94	0.64–1.40	0.769	0.86	0.58–1.29	0.476
2–3	1.50	0.97–2.32	0.068	1.26	0.80–1.99	0.315
White vs. non*–*white	0.70	0.48*–*1.01	0.055	0.72	0.49*–*1.05	0.089
Academic vs. non*–*academic	1.10	0.85*–*1.44	0.460	*––––*	*––––*	*––––*
Uninsured vs. insured	1.11	0.55*–*2.21	0.775	*––––*	*––––*	*––––*
Median income (<$46 K/yr vs. ≥$46 K/yr)	1.21	0.94*–*1.55	0.146	1.15	0.89*–*1.49	0.276
Tumor Size						
≤1 cm	Ref	Ref	Ref	Ref	Ref	Ref
1–2 cm	0.71	0.53–0.96	0.024	0.69	0.51–0.93	0.014
Unknown	1.07	0.77–1.48	0.683	1.01	0.72–1.40	0.975
Distance (per mile)	1.00	1.00*–*1.00	0.544	*––––*	*––––*	*––––*

CRT: chemoradiation. RT: radiation therapy. OR > 1 implies that RT alone is more likely while OR < 1 implies that CRT is more likely.

**Table 3 cancers-12-03248-t003:** Univariate and multivariate Cox regression analysis for overall survival.

Variable	Univariate Analysis			Multivariate Analysis		
	OR	95% CI	*p*-Value	OR	95% CI	*p*-Value
CRT vs. RT	0.54	0.43–0.68	<0.001	0.65	0.52–0.82	<0.001
Age (years)						
<50	1.21	0.94–1.56	0.148	0.96	0.74–1.25	0.757
50–59	Ref	Ref	Ref	Ref	Ref	Ref
60–69	1.57	1.25–1.96	<0.001	1.53	1.22–1.91	<0.001
≥70	3.89	3.17–4.76	<0.001	3.65	2.97–4.48	<0.001
Female vs. male	0.65	0.55*–*0.75	<0.001	0.69	0.59*–*0.81	<0.001
Charlson–Deyo score						
0	Ref	Ref	Ref	Ref	Ref	Ref
1	1.82	1.49–2.23	<0.001	1.50	1.22–1.84	<0.001
2–3	2.99	2.35–3.81	<0.001	2.77	2.15–3.56	<0.001
White vs. non*–*white	0.81	0.64*–*1.04	0.099	0.84	0.65*–*1.07	0.317
Academic vs. non*–*academic	0.82	0.69*–*0.98	0.025	0.85	0.72*–*1.02	0.065
Uninsured vs. insured	1.01	0.66*–*1.54	0.982	*––––*	*––––*	*––––*
Median income (<$46 K/yr vs. ≥$46 K/yr)	1.41	1.21*–*1.66	<0.001	1.27	1.08*–*1.50	0.003
Tumor Size ≤1 cm 1*–*2 cm Unknown	Ref 0.96 1.29	Ref 0.79*–*1.16 1.04*–*1.58	Ref 0.669 0.018	Ref 0.91 1.16	Ref 0.75*–*1.10 0.95*–*1.43	Ref 0.322 0.154
Distance (per mile)	1.00	1.00*–*1.00	0.471	*––––*	*––––*	*––––*

CRT: chemoradiation. RT: radiation therapy.

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
