# Peer review of "Stage I Squamous Cell Carcinoma of the Anus: Is Radiation Therapy Alone Sufficient Treatment?"

_cancers, 2020, doi:10.3390/cancers12113248_

Round 1

Reviewer 1 Report

Dear authors, I want to express my compliments about your manuscript.

The manuscript is well done in term of reaserch and statistical design. As mentioned the number of patients is large to have conclusions about the end point of the study. Limitations are well described (retrospective study, lack of data about local reccurence, cancer specific survival, details about chemotherapy)

In my opinion, may be it will be better to change the form of manuscript like as materials and methods before results.

Author Response

REVIEWER 1

Dear authors, I want to express my compliments about your manuscript.

The manuscript is well done in term of reaserch and statistical design. As mentioned the number of patients is large to have conclusions about the end point of the study. Limitations are well described (retrospective study, lack of data about local reccurence, cancer specific survival, details about chemotherapy)

Response: We thank the reviewer for these comments.

In my opinion, may be it will be better to change the form of manuscript like as materials and methods before results.

Response: We thank the reviewer for this suggestion. As per the journal editorial guidelines, the Materials and Methods comes at the end of the manuscript.

Reviewer 2 Report

Cancers 962002

In their well-written manuscript entitled “Stage I Squamous Cell Carcinoma of the Anus: Is Radiation Therapy Alone Sufficient Treatment?” Miller and colleagues address the question of best treatment of patients with Stage I cancers who were not well-represented on trials demonstrating a benefit of concurrent chemotherapy. Given that there is data suggesting excellent disease-specific outcomes for radiation alone (Myerson, et al, refs 8-10), and that the chemotherapy used for this disease (typically MMC + 5FU) is associated with significant toxicity as the authors point out, this is a very important question. The authors utilize the NCDB to compare overall survival of patients with Stage I SCCA who received chemotherapy and radiation or radiation alone, with propensity matching methods. They conclude that since OS was higher in the CRT cohort, that this should represent the standard treatment for these patients. While such a query of the NCDB would be valuable to include in the literature, the following concerns/comments are raised, with suggestions to improve the impact of the analysis.

Major points:

  1. The authors are commended for limiting the study population to those receiving standard RT doses, those with known RT or chemo details, and those with SCC histology. Despite only requiring follow up of >=1month from diagnosis (which is not sufficient), the actual f/u IQR was sufficient.
  2. One thing is clear from this study – the large majority of patients with Stage I SCCA are being treated with CRT. Access to treatment (indicated by insurance coverage, distance from facility, and facility type), on the other hand, was not different between the unmatched cohorts. Tumor size was larger in the CRT arm, but with 25% unreported values. Thus, there must be some reason, whether documented or not, that certain patients are not being treated with chemotherapy. The assumption is that these patients are not well enough to receive chemotherapy, which is supported by the current data – patients who were treated with RT only were older, and more often had multiple comorbidities. Limitations of Charlson-Deyo comorbidity index are well-known and this remains a major limitation to the impact of these findings. This major limitation should be expanded upon in the discussion.
  3. Charlson-Deyo is reported in NCDB as 0, 1, 2 or 3. Were there no patients in the original >3,800 patients with stage I disease who had CD score of 3 (which accounts for >2 points on the index)?
  4. Can the authors comment on how OS for this cohort relates to cancer-specific survival? CSS would be expected to be >90% long-term for stage I anal SCC, from retrospective series including all treatment approaches. Thus it is unclear what the OS endpoint indicates, other than imbalanced co-morbidities. To this point, the difference in survival at 3- and 5-years in ACT I and EORTC randomized trials were not significant and were not as great a magnitude as suggested in this analysis.  Can the authors evaluate other potential surrogates for disease recurrence – RT, surgery or chemo delivered long after initial treatment? “additional treatment,” “palliative care,” or “immunotherapy” (which would only be administered in the setting of recurrence)?
  5. The propensity score matching was performed using a standard and common approach; however, dichotomized variables for age (cut off 65) and Charlson-Deyo score (>=1 vs 0) are very limiting in the context of this study. The analysis should be repeated with matched age groups (5 year intervals, for instance), and exact matched comorbidity index, or an explanation of why this is not possible should be provided. The nearest-neighbor match should be employed with the exact numbers for age interval and C-D score, or again, a reason for not doing this provided. This is especially important for age given the high HR on MVA for RT only.
  6. The fact that OS in the WLE vs CRT NCDB study was not different but there was a difference in this NCDB study suggests a strong retrospective bias in this study that cannot be overcome by propensity score matching. Patients eligible for surgery would presumably have fewer unaccounted co-morbidities or other unmeasured factors than patients not receiving CRT. Can the authors recreate the WLE cohort in the NCDB as a third PS matched group to show that this sub-group patients also did as well as those with CRT? Or would those patients also have inferior OS compared to CRT group? If this were true, it would strengthen the argument that RT alone may not be sufficient treatment for these patients.

Minor points:

  1. The conclusion at the end of the summary should be softened.
  2. Duration of radiation therapy (days) should be reported and if uneven should be included in the propensity score matching algorithm.

References:

Myerson et al, Int. J. Radiation Oncology Biol. Phys., Vol. 75, No. 2, pp. 428–435, 2009.

Author Response

In their well-written manuscript entitled “Stage I Squamous Cell Carcinoma of the Anus: Is Radiation Therapy Alone Sufficient Treatment?” Miller and colleagues address the question of best treatment of patients with Stage I cancers who were not well-represented on trials demonstrating a benefit of concurrent chemotherapy. Given that there is data suggesting excellent disease-specific outcomes for radiation alone (Myerson, et al, refs 8-10), and that the chemotherapy used for this disease (typically MMC + 5FU) is associated with significant toxicity as the authors point out, this is a very important question. The authors utilize the NCDB to compare overall survival of patients with Stage I SCCA who received chemotherapy and radiation or radiation alone, with propensity matching methods. They conclude that since OS was higher in the CRT cohort, that this should represent the standard treatment for these patients. While such a query of the NCDB would be valuable to include in the literature, the following concerns/comments are raised, with suggestions to improve the impact of the analysis.

Major points:

  1. The authors are commended for limiting the study population to those receiving standard RT doses, those with known RT or chemo details, and those with SCC histology. Despite only requiring follow up of >=1month from diagnosis (which is not sufficient), the actual f/u IQR was sufficient.

Response: We thank the reviewer for this comment.

  1. One thing is clear from this study – the large majority of patients with Stage I SCCA are being treated with CRT. Access to treatment (indicated by insurance coverage, distance from facility, and facility type), on the other hand, was not different between the unmatched cohorts. Tumor size was larger in the CRT arm, but with 25% unreported values. Thus, there must be some reason, whether documented or not, that certain patients are not being treated with chemotherapy. The assumption is that these patients are not well enough to receive chemotherapy, which is supported by the current data – patients who were treated with RT only were older, and more often had multiple comorbidities. Limitations of Charlson-Deyo comorbidity index are well-known and this remains a major limitation to the impact of these findings. This major limitation should be expanded upon in the discussion.

Response: We agree with the reviewer on this point. We have expanded on the limitations of not knowing performance status in the Discussion on page 9, lines 247-249.

  1. Charlson-Deyo is reported in NCDB as 0, 1, 2 or 3. Were there no patients in the original >3,800 patients with stage I disease who had CD score of 3 (which accounts for >2 points on the index)?

Response: In the initial version of the manuscript, we chose to dichotomize the Charlson-Deyo score at 0 since 82.6% of patients had a score of 0 and only 17.4% combined had scores of 1 (11.3%), 2 (2.2%), or 3 (3.9%). In this version of the manuscript, we categorize the Charlson-Deyo Score into 3 groups: 0 (82.6%), 1 (11.3%) and 2-3 (6.1%). This categorization has been updated in all of the tables/analyses.

  1. Can the authors comment on how OS for this cohort relates to cancer-specific survival? CSS would be expected to be >90% long-term for stage I anal SCC, from retrospective series including all treatment approaches. Thus it is unclear what the OS endpoint indicates, other than imbalanced co-morbidities. To this point, the difference in survival at 3- and 5-years in ACT I and EORTC randomized trials were not significant and were not as great a magnitude as suggested in this analysis.  Can the authors evaluate other potential surrogates for disease recurrence – RT, surgery or chemo delivered long after initial treatment? “additional treatment,” “palliative care,” or “immunotherapy” (which would only be administered in the setting of recurrence)?

Response: We agree that a major limitation of the NCDB is that the only outcome measure captured is Overall Survival. There are no data on cancer-specific survival, distant metastases, local recurrences, etc. We have included your point that the magnitude in the survival difference is greater than that seen in ACT1 and EORTC in the Discussion on page 7-8, lines 185-187. As you point out, other unmeasured confounders, such as performance status, could be influencing the OS results [this is now mentioned in the Discussion on p.8, lines 187-190]. Last, the NCDB captures treatment information only regarding the first disease presentation. Treatments that were rendered at the time of disease recurrence/progression are not captured. We have added this to the Discussion on p.8, lines 190-192.

  1. The propensity score matching was performed using a standard and common approach; however, dichotomized variables for age (cut off 65) and Charlson-Deyo score (>=1 vs 0) are very limiting in the context of this study. The analysis should be repeated with matched age groups (5 year intervals, for instance), and exact matched comorbidity index, or an explanation of why this is not possible should be provided. The nearest-neighbor match should be employed with the exact numbers for age interval and C-D score, or again, a reason for not doing this provided. This is especially important for age given the high HR on MVA for RT only.

Response: We thank the reviewer for this point. We have now categorized age into 4 groups: <50 years old, 50-59 years old, 60-69 years old, and 70 or older. In addition, the Charlson-Deyo Score is now broken down into 3 groups: 0, 1, and 2-3. Our propensity score matched analysis is now based on age and Charlson-Deyo score using these categories. We presented updated pre- and post-match results using these categories.

  1. The fact that OS in the WLE vs CRT NCDB study was not different but there was a difference in this NCDB study suggests a strong retrospective bias in this study that cannot be overcome by propensity score matching. Patients eligible for surgery would presumably have fewer unaccounted co-morbidities or other unmeasured factors than patients not receiving CRT. Can the authors recreate the WLE cohort in the NCDB as a third PS matched group to show that this sub-group patients also did as well as those with CRT? Or would those patients also have inferior OS compared to CRT group? If this were true, it would strengthen the argument that RT alone may not be sufficient treatment for these patients.

Response: While the initial aim of this manuscript was to focus just on patients that received chemoradiation or RT alone, we recognize that wide local excision is a common treatment in stage I anal cancer. In our propensity-score matched analysis of CRT patients versus those treated with excision alone, we found that there was a slightly higher risk of death with WLE compared to CRT (82.8% vs. 85.6%, HR=1.20 (95% CI 1.00-1.43, p=0.0450). This slight difference in favor of CRT may be due to our strict definitions of CRT, which were not reported in the previous study, and also to the large sample size. As point out by the reviewer, this also argues that CRT may be the preferred treatment for stage I patients. We have added this to the Results on page 5, lines 121-126 and to the Discussion on page 8, lines 192-199.

Minor points:

  1. The conclusion at the end of the summary should be softened.

Response: We agree. The conclusion has been softened.

  1. Duration of radiation therapy (days) should be reported and if uneven should be included in the propensity score matching algorithm.

Response: Thank you for bringing up this point. The duration of RT was similar in the RT alone vs. CRT groups (51.5 days [80.2] vs. 54.4 days [95.0], p=0.6147]. We have added this to the Results section on page 2, lines 87-88.

References:

Myerson et al, Int. J. Radiation Oncology Biol. Phys., Vol. 75, No. 2, pp. 428–435, 2009.

Reviewer 3 Report

The authors report on stage I anal cancer patients included in the NCD (between 200154 and 2015) and receiving exclusive RT or combined modality treatment with concurrent chemo-radiation. The authors used propensity-score matched analysis to demonstrate an advantage in terms of overall survival for patients treated with concurrent RT-CT. The topic is of interest and the manuscript is clear, well-written and based on a robust methodology. I do have comments:

  • Authors state that T1N0 pts were underrepresented in the UKCCR ACT I trial and hence the aforementioned trial may not be informative for this setting of patients. This is partially true.

In the ACT I trial, investigators included 71 patients with T1 tumours. I agree that the numbers are small, but there is  a letter you may not be aware of in the Lancet.
Northover J, Meadows H, Ryan C, Gray R on behalf of UKCCCR Anal Cancer Trial Working Party. Combined radiotherapy and chemotherapy for anal cancer. The Lancet 1997; 349: 9046:205-206. The letters says:

“In Act I there were 223 patients with T1-2,N0 stage. There  was  a clearly  significant  advantage  for  CMT in avoidance of local failure in both T1N0 (p=0·0352,  RR  0·35,  95%  CI 0·12–0·97) and T2N0 (p=0·0049, RR 0·49,  95%  CI  0·29–0·81)  (figure).Taken   together,   this   early   group yielded    the    following    statistical advantage for  CMT:  p=0·0005,  RR 0·45,  95%  CI  0·29–0·71.”

            Hereby  slide showing RR for local recurrence according to stage in ACT I

You may want to elaborate more on this point.

  • With respect to the added toxicity due to chemotherapy you mentioned, you may want to cite and discuss the paper by Franco et al. (Early stage node negative (T1-T2N0) anal cancer treated with simultaneous integrated boost radiotherapy and concurrent chemotherapy. Anticancer Res 2016;36:1943-8), where early stage anal cancer pts were treated with chemoradiation, delivered with IMRT and IGRT and a SIB approach. The toxicity profile was favourable. You may want to discuss how the RT approach may influence compliance and toxicity in this setting.
  • With respect to de-escalation strategies in this setting: the authors focus on the omission of CT as a de-escalation strategies. This is not the only way to de-escalate: you may want to cite for example the omission of MMC suring the second cycle of CT (see White et al, Radiother Oncol 2015) or the use of capecitabine (Goodman et al, IJROBP 2017). Another way to de-escalate (keeping CT on) is act on radiation volumes (omission of inguinal irradiation in properly staged pts) and decrease in RT dose. I would suggest to discuss about RT de-escalation, particularly in light of your findings, which seem to confirm the need to keep CT on board for these patients.
  • In the setting of de-escalation it is hard not to cite and discuss the PLATO trials (Personalizing anal cancer radiotherapy dose), in particular ACT 3 and 4 which are presently running in the UK.

Author Response

The authors report on stage I anal cancer patients included in the NCD (between 200154 and 2015) and receiving exclusive RT or combined modality treatment with concurrent chemo-radiation. The authors used propensity-score matched analysis to demonstrate an advantage in terms of overall survival for patients treated with concurrent RT-CT. The topic is of interest and the manuscript is clear, well-written and based on a robust methodology. I do have comments:

Authors state that T1N0 pts were underrepresented in the UKCCR ACT I trial and hence the aforementioned trial may not be informative for this setting of patients. This is partially true.

In the ACT I trial, investigators included 71 patients with T1 tumours. I agree that the numbers are small, but there is  a letter you may not be aware of in the Lancet.
Northover J, Meadows H, Ryan C, Gray R on behalf of UKCCCR Anal Cancer Trial Working Party. Combined radiotherapy and chemotherapy for anal cancer. The Lancet 1997; 349: 9046:205-206. The letters says:

“In Act I there were 223 patients with T1-2,N0 stage. There  was  a clearly  significant  advantage  for  CMT in avoidance of local failure in both T1N0 (p=0·0352,  RR  0·35,  95%  CI 0·12–0·97) and T2N0 (p=0·0049, RR 0·49,  95%  CI  0·29–0·81)  (figure).Taken   together,   this   early   group yielded    the    following    statistical advantage for  CMT:  p=0·0005,  RR 0·45,  95%  CI  0·29–0·71.”

            Hereby  slide showing RR for local recurrence according to stage in ACT I

You may want to elaborate more on this point.

Response: We thank the reviewer for bringing up this point and including the slide. We have now incorporated this into the Introduction on page 2, lines 66-68 and the Discussion on page 7, lines 146-148.

With respect to the added toxicity due to chemotherapy you mentioned, you may want to cite and discuss the paper by Franco et al. (Early stage node negative (T1-T2N0) anal cancer treated with simultaneous integrated boost radiotherapy and concurrent chemotherapy. Anticancer Res 2016;36:1943-8), where early stage anal cancer pts were treated with chemoradiation, delivered with IMRT and IGRT and a SIB approach. The toxicity profile was favourable. You may want to discuss how the RT approach may influence compliance and toxicity in this setting.

Response: We again thank the reviewer for this excellent suggestion. We now include this in the Discussion on page 8, lines 212-218.

With respect to de-escalation strategies in this setting: the authors focus on the omission of CT as a de-escalation strategies. This is not the only way to de-escalate: you may want to cite for example the omission of MMC suring the second cycle of CT (see White et al, Radiother Oncol 2015) or the use of capecitabine (Goodman et al, IJROBP 2017). Another way to de-escalate (keeping CT on) is act on radiation volumes (omission of inguinal irradiation in properly staged pts) and decrease in RT dose. I would suggest to discuss about RT de-escalation, particularly in light of your findings, which seem to confirm the need to keep CT on board for these patients.

Response: We thank the reviewer for these great points. We have now included a more detailed discussion on treatment de-escalation including radiation de-escalation in the Discussion on page 8, lines 220-234.

In the setting of de-escalation it is hard not to cite and discuss the PLATO trials (Personalizing anal cancer radiotherapy dose), in particular ACT 3 and 4 which are presently running in the UK.

Response: We again thank this reviewer for the suggestion. We now have including the PLATO trials in our Discussion on pages 8-9, lines 237-241.

Round 2

Reviewer 2 Report

All of the reviewer's concerns have been addressed in the revised manuscript. Remaining limitations are related to the database and are addressed in the discussion.